# Caspian Sea Mycosands: The Variety and Abundance of Medically Important Fungi in Beach Sand and Water

**DOI:** 10.3390/ijerph20010459

**Published:** 2022-12-27

**Authors:** Maryam Moazeni, Mohammad Taghi Hedayati, Iman Haghani, Mahdi Abastabar, Abolfazl Saravani Jahantigh, Maryam Kheshteh, Mojtaba Nabili, João Brandão

**Affiliations:** 1Invasive Fungi Research Center, Communicable Diseases Institute, Mazandaran University of Medical Sciences, Sari 48157-33971, Iran; 2Department of Medical Mycology, School of Medicine, Mazandaran University of Medical Sciences, Sari 48157-33971, Iran; 3Student Research Committee, Mazandaran University of Medical Sciences, Sari 48157-33971, Iran; 4Department of Medical Laboratory Sciences, Faculty of Medicine, Sari Branch, Islamic Azad University, Sari 48161-19318, Iran; 5Departamento de Saúde Ambiental, Instituto Nacional de Saúde Doutor Ricardo Jorge, Avenida Padre Cruz, 1649-016 Lisbon, Portugal; 6Centre for Environmental and Marine Studies (CESAM), Department of Animal Biology, University of Lisbon, 1649-004 Lisbon, Portugal

**Keywords:** sand, Flavi, water quality, regulation

## Abstract

Samples from a total of 67 stations, distributed amongst 32 cities along the Caspian Sea coastline, were collected during the summer of 2021 on sunny days. The samples were collected from each station, including both dry/wet sand and shoreline water. The grown samples were primarily analyzed for the macro/microscopic morphologic features of the fungi. Moreover, identification by PCR-RFLP was performed for yeasts, dermatophytes, and *Aspergillus* sp. strains. Antifungal susceptibility tests were performed for probable-isolated *Aspergillus* and *Candida* sp. A total of 268 samples were collected, from which 181 (67.54%) isolates were recovered. Yeast-like fungi and potential pathogenic black fungi were detected in 12 (6.6%) and 20 (11%) of the sand (dry/wet) samples. Potential pathogenic hyaline fungi were identified in 136 (75.1%) samples, in which *Aspergillus* sp. was the predominant genus and was detected in 76/136 (47.8%) samples as follows: *A*. section *Flavi n* = 44/76 (57.9%), *A*. section *Nigri n* = 19/76 (25%), *A*. section *Nidulantes n* = 9/76 (11.8%), and *A*. section *Fumigati n* = 4/76 (5.3%). The most effective azole antifungal agent was different per section: in *A*. section *Fumigati*, PSZ; in *Aspergillus* section *Nigri*, ITZ and ISZ; in *A*. section *Flavi*, EFZ; and in *A*. section *Nidulantes*, ISZ. *Candida* isolates were susceptible to the antifungals tested.

## 1. Introduction

Marine and coastal environments represent transitional ecosystems that connect terrestrial and marine locations. The Caspian Sea is considered as the largest closed water body on the surface of the Earth and its isolation makes it a very unique ecosystem [1]. Five countries, namely Kazakhstan, Turkmenistan, Azerbaijan, Russia, and Iran, are located along roughly 4800 km (3000 mi) of the Caspian coastline. The length of the coastline located in Iran is 728 km (452 mi), covering three provinces, comprising Mazandaran, Guilan, and Golestan (https://en.wikipedia.org/wiki/Caspian_Sea, accessed on 25 October 2022). The Caspian Sea’s beaches and shorelines are highly appreciated areas for recreation in Iran and are responsible for a significant portion of its tourism industry income. Rating these areas for public health use is of extreme importance for the coastal communities. In other words, an infection outbreak associated with beaches will have serious negative consequences regarding economic and social impacts on the surrounding areas. Hence, recreational water quality needs to be monitored to evaluate the threat of water-borne illnesses [2]. Recently, environmental pollution has become an increasing problem in this region. The Caspian Sea is heavily polluted due to industrial and agricultural effluents, as well as the extraction of oil and gas reserves [1]. Based on the new World Health Organization (WHO) guidelines for recreational water quality assessment, the traditional fecal indicator parameter, the enterococci enumeration per volume of water, is still key [3]. The European bathing water directive (BWD), however, sets tighter standards by requiring the establishment of a “bathing water profile”, which results in the “identification and assessment of causes of pollution that might affect bathing waters and impair bathers’ health”, and is now under review [4]. WHO recently pointed out the fungal priority pathogens list to guide research, development, and public health action [5], but while fungal species are now considered in the WHO guidelines, unfortunately, they are not considered in the current BWD recommendations. However, there are opportunistic pathogens directly influencing human health, especially among those who are vulnerable due to underlying medical conditions, such as allergies, pneumonia, bronchitis, diabetes, or immune suppression [2]. Most fungi are frequently found in sand and survive longer than other microorganisms due to the presence of spores. It was observed that many fungi, such as *Candida albicans* and some species of dermatophytes (*Trichophyton rubrum*, *T. mentagrophytes*, *Microsporum gypseum*, *M. canis, Epidermophyton floccosum*), were capable to remain alive in non-sterile sand for 30 to 360 days [2]. Accordingly, some researchers suggested that pathogenic yeast-like fungi found in beach sand could be a good mycological indicator in evaluating the safety of marine waters [6]. It is highly expected that beachgoers constantly exposed to sand containing different types of fungal species are at an increased risk through direct contact with their skin and mucous membranes or by inhaling spores. Nevertheless, no association has been reported between pathogenic fungi and related infections in beach sands so far [7]. In addition, the reverse issue may be possible as both humans and animals may themselves be partially contributing to beach sand microorganisms [7]. In 2009, Heaney et al. performed an epidemiological study on beachgoers and reported a strong correlation between sand contact and enteric illness at marine beaches [8]. In 2020, Brandão et al. published an actual case of a sand-borne outbreak in 30 young beach users, caused by a raw sewage spill from a beach bar located on a cliff [9]. Fungal genera that have been isolated from beach sands include *Aspergillus*, *Chrysosporium*, *Fusarium* [10], *Scedosporium*, *Scytalidium*, *Scopulariopsis* [11], *Candida* [12], *Penicillium*, *Rhodotorula* [6], *Cladosporium*, *Mucor*, *Stachybotrys* [13,14,15,16], *Phialemonium* [17], and many others [18]. *Trichophyton* and *Microsporum*, associated with skin and nail infections, also have been isolated from beach sand [11]. The present work reports on the mycological quality of the sand at selected beaches, including the variety and abundance of the species, as well as the antifungal susceptibility profiles of fungi collected from Caspian Sea beach sands.

## 2. Materials and Methods

### 2.1. Study Design

According to geographical borders, the Caspian Sea coastline in Iran was divided into three areas (Mazandaran, Guilan, and Golestan), which included a total of 67 stations, and they were analyzed during summer 2021. Based on the unique tourism situation and the population density of beach users, a number of beach stations was considered in each city.

### 2.2. Sample Collection

Four samples were collected from each station, including dry sand collected from the middle of the dry sand section of the shore (20 cm in depth and 2 m away from the shoreline); wet sand comprising coastline areas (one meter away from the waterline) and also the shallow part (20 cm in depth) of seawater, i.e., approximately one meter towards the sea water; and, finally, a water sample that was collected from water 3 m away from the shoreline (50 cm in depth). The sampling event occurred at the peak of the summer (highest concentration of users) from 1 June to 15 September 2021. The samples were collected only on sunny days between 11 a.m. and 15 p.m., when there were slight temperature changes. In total, 10 g sand and 10 mL water samples were collected with sterile gloves into a sterile plastic container. Each sand sample was collected by combining the sub-samples from the corners and the center of a 2 × 2 square. Sub-samples were then combined and a total of 40 g of sand sample per each site was transported to the laboratory and processed within 24 h.

### 2.3. Detection and Identification

Each wet sand sample, either from the coastline or from the shallow part of sea water, (10 g—not oven-dried prior to processing, maintaining its natural water content), was agitated for 30 min at 100 rpm, and 0.2 mL of this suspension was spread onto Sabouraud Dextrose Agar petri dishes supplied with Cyclohexamide (Sigma-Aldrich, St. Louis, MO, USA) (0.5 g/L) and chloramphenicol (0.05 g/L) (for dermatophyte fungi), as well as Saubouroud Dextrose Agar (Merck, Darmstadt, HE, Germany) with only chloramphenicol (0.05 g/L) (for non-dermatophyte fungi), and incubated for up to 3 weeks at 28 °C and 5–7 days at 30 °C for dermatophytes and non-dermatophyte fungi, respectively. Dry sand samples were mixed for a few minutes and then 2 g of the whole sample was placed onto the same petri dishes as mentioned above. In the case of water samples, 0.2 mL water samples were also cultured as above after mixing for a few minutes. Yeasts were preliminarily identified using CHROMagar *Candida* medium (CHROMagar Microbiology, France). Mold fungal identification was also initially carried out via macroscopic (color, texture, color behind the colony, and the presence of pigment) and microscopic features (macroconidia and microconidia, their shape, and their appearance) using lactophenol blue staining [19,20]. The PCR-RFLP method was designed for the molecular identification of probable isolates of yeast, dermatophytes, and *Aspergillus* species. To achieve this, DNA extraction was performed using the previously published protocols. Briefly, a small number of hyphae/yeast cells from young colonies were transferred to Eppendorf tubes containing 300 μL of lysis buffer (200 mM Tris–HCl; pH 7.5), 25 mM ethylenediaminetetraacetic acid (EDTA), 0.5% *w/v* sodium dodecyl Sulfate (SDS) (Merck, Darmstadt, HE, Germany), 250 mM NaCl), and were mechanically crumpled with 300 glass beads (0.5 mm). Fungal DNA was then extracted using the phenol–chloroform assay. Sodium acetate and isopropanol were applied for DNA precipitation, and finally the DNA pellets were washed with cold 70% ethanol and dried under a heat block [21,22,23]. Table 1 shows the sequences of applied primers for the molecular evaluation of the isolates’ species. Isolated strains were identified to the species level by amplifying and digesting the specific gene/region, including the internal transcribed spacer (ITS1-5.8s-ITS2) for *Candida* and dermatophytes, using *Msp*1 and *Mva*1 restriction enzymes, respectively. The species that were identified as *Aspergillus* were then analyzed by digesting the β-tubulin gene with *Alw*I (*BspP*I) (LifeTechnologies, Carlsbad, CA, USA) restriction enzyme [21,23,24,25].

### 2.4. Antifungal Susceptibility Tests

Antifungal susceptibility tests were performed using the Clinical and Laboratory Standards Institute (CLSI) guidelines M38, M60, and M59 for probable isolated *Aspergillus* and yeast species, respectively [26,27,28]. Due to the lack of financial resources, AFSTs were performed for the most medically important fungi, such as isolates belonging to the genera *Aspergillus* and *Candida*. The antifungal agents were diluted in a standard RPMI 1640 medium (Sigma-aldrich, St. Louis, MO, USA), and then buffered to pH 7.0 with 0.165 M 3-(N-Morpholino) propanesulfonic acid (MOPS, Sigma Chemical Co.) with L-glutamine without bicarbonate to yield two times their concentration. Subsequently, they were distributed into 96-well microdilution trays (Nunc, UK) with a final concentration of 0.016–16 μg/mL for Itraconazole (ITZ), Voriconazole (VRZ), Posaconazole (PSZ), Isavuconazole (ISZ), Efinaconazole (EFZ), Miconazole (MYZ), and Amphotericin B (AMB), regarding 0.063–64 μg/mL for Fluconazole. MICs were read visually at 100% inhibition of growth after 24 h of incubation at 35 °C for tested drugs compared to positive controls. *Candida parapsilosis* (ATCC 22019) and *Hamigera insecticola* (previously identified as *Paecilomyces variotii*) (ATCC 22319) were used as the quality control isolates and were included on each day of testing.

### 2.5. Statistical Analyses

All statistical analyses were performed using SPSS v25 (SPSS Inc., Chicago, IL, USA). Descriptive analysis for parameters was performed, using means and 95% confidence intervals for mean, median, minimum, and maximum values for continuous variables, and Student’s t-test was performed on all variables of this study. MIC ranges, MIC_50S_, MIC_90S_, and geometric mean (GM) MICs were calculated. Statistical significance was assumed with a *p* value of ≤0.05.

## 3. Results

### 3.1. Species Identification

A total of 268 samples from 67 stations were gathered and admitted to medical mycology laboratories at the Invasive Fungi Research Centre (IFRC), Sari, Iran. The distribution of the evaluated stations was as follows: Mazandaran province—41 (61.2%), with 164 collected samples; Guilan province—20 (29.8%), with 80 collected samples; and Golestan province—6 (8.9%), with 24 collected samples. Of the 268 analyzed samples, 181 (67.54%) were positive for the fungal species, of which 100 (55.25%) strains, including 92 filamentous fungal species and 8 yeast species, were isolated from Mazandaran province. Fifty-five (51 filamentous fungi and 4 yeast species) (30.4%) and 26 (all filamentous fungi) (14.4%) isolates were collected from Guilan and Golestan provinces, respectively. Yeasts and yeast-like fungi were detected in 12 (6.6%) of the sand (dry/wet) samples. Of the detected yeasts, the following species were identified: *C. albicans*—*n* = 3, *C. tropicalis*—*n* = 2, other yeast species—*n* = 4, and *Trichosporon* sp.—*n* = 3. Potential pathogenic black fungi were found in 20 (11%) of the samples (Table 2) with the following distribution: *Bipolaris* sp.—*n* = 2 (1.1%), *Cladosporium* sp.—*n* = 5 (2.8%), *Alternaria* sp.—*n* = 3 (1.6%), and other dematiaceous fungal species—*n* = 10 (5.5%). Potential pathogenic hyaline fungi were identified in 136 (75.1%) samples. Among these, *Aspergillus* sp. was the predominant genus, detected in 76/136 (47.8%) samples with the following distribution: *A.* section *Flavi*—*n* = 44/76 (57.9%), *A.* section *Nigri*—*n* = 19/76 (25%), *A.* section *Nidulantes*—*n* = 9/76 (11.8%), *A.* section *Fumigati*—*n* = 4/76 (5.3%). The distribution of other hyaline fungal species was identified as follows: *Penicillium* sp.—*n* = 21 (11.6%), *Fusarium* sp.—*n*= 7 (3.9%), *Trichoderma* sp.—*n* = 18 (9.9%), *Rhizopus* sp.—*n* = 4 (2.2%), *Mucor* sp.—*n* = 5 (2.8%), *Geotrichum* sp.—*n* = 2 (1.1%), *Acromonium*—*n* = 3 (1.6%). Table 2 lists the identified fungal isolates in detail. Regarding dermatophytes, they were found in 13 (7.2%) of the examined samples and the most prevalent genus comprised species belonging to the *T. mentagrophytes/interdigitale* species complex, detected in 10 sand (dry and wet) samples. *M. canis* was detected in one and *M. gypseum* in two samples, whereas no *Epidermophyton* isolates were recovered. Figure 1 indicates the distribution of identified fungal species isolated from the whole Caspian Sea coastline regarding provinces. A large variety of fungal species were reported from sea beach sand samples (dry as well as wet) rather than water. Among these, *A.* section *Flavi* was the predominant species isolated from different types of samples. A significant increase in the isolates collected from sand was observed when a comparison was made among the number of strains from both sand and water (*p* < 0.05). However, no significant differences were observed among the number of collected isolates between dry and wet sand. In addition, no notable differences were shown between water and water sand samples (*p* > 0.05). In total, 133 samples, including 72 dry sand samples and 61 wet sand samples, and a total of 48 water samples, including 21 water sand and 27 water samples, were confirmed as positive fungus-growing strains.

### 3.2. Susceptibility Profile

Regarding M59 [28], 10.5% (2/19 isolates) and 11.4% (5/44 isolates) of *A.* section *Nigri* and *A.* section *Flavi*, and one isolate from *A.* section *Fumigati*, were non-wild types against PSZ, respectively. In total, 10.5% (8/76 strains) of all *Aspergillus* strains were not wild type and showed high MICs against PSZ. In the case of ITZ, VRZ, and ISZ, only 2.6% (one strain of *A.* section *Flavi* and one strains of *A.* section *Fumigati*), 2.6% (one strain belonging to *A.* section *Nigri* and one from *A.* section *Fumigati*), and 5.3% (one from *A.* section *Flavi*, two strains of *Aspergillus* section *Nigri*, and one from *A.* section *Fumigati*) of the total *Aspergillus* isolates were reported as non-wild types, respectively. Table 3 indicates the detailed parameters of AFST. Accordingly, among azole antifungals, the most effective antifungal agent was different within each section and was reported as follows: in *A.* section *Fumigati*, PSZ; in *A.* section *Nigri*, ITZ and ISZ; in *A.* section *Flavi*, EFZ; and in *A.* section *Nidulantes*, ISZ. Based on the documented guidelines [27,28], all *Candida* isolates were wild types/susceptible against the examined antifungal agents. According to the GM values, Isavaconazole was the most effective agent against *Candida* strains.

## 4. Discussion

Beaches can be considered as a passive source of pollution that may be contaminated with animal or human waste or garbage. Even water may carry pathogenic microorganisms, including fungi [29]. Moreover, both sand beaches and water should be considered as a source carrying emerging pathogenic fungal species that could be harmful for some human users. There is no epidemiological evidence or Quantitative Microbial Risk Assessment (QMRA) calculation of the potential of human fungal infections and fungal contamination starting at the beach by exposure to sand beaches; however, beach users suffering from immunosuppression disorders, at any level, are more vulnerable to infection with microorganisms such as fungi [30,31]. Moreover, WHO recently published a priority watchlist of fungal taxa, critically recommending the monitorization of most taxa addressed in this text [5]. Apart from the behavior of beachgoers and bathers, which can clearly affect the sand mycobiome, different types of species may be introduced to the sand coastline by water movements from the depth of the sea to the wet or dry part of the shore [32]. Additionally, the presence of pets as well as a huge number of beach users can affect and change the distribution of fungal species. In Iran, pets are not usually taken to the beach; if so, dogs are considered the most common. In addition, some recreational activities, such as horse-riding, are common in Iran and can affect the mycobiome of sand and also water. Although this issue was not considered in this study, WHO recommends that pets are taken to non-bathing areas instead of designated beaches [3]. Hence, restricting the above factors will reduce sand pollution and eventually human exposure to various pathogenic fungi [33]. The native microbiota of the sand beaches may change due to microbial deposition, coast retraction, and emerging antimicrobial resistance [34]. In addition, global warming, the human population, and climate change are expected to also affect the diversity and abundance of microbiota, including mycobiota [18]. For the first time in Iran, this study provides data on the fungal content of sand and water along the Caspian Sea coast during summer 2021. Several other researchers also showed that sand contamination during the summer was higher than during spring [29,35]. Accordingly, and also due to the unique tourism appeal of northern beaches in Iran, the Caspian Sea coastlines are very crowded during the summer, which was why the summer was selected for this study. Results from several studies showed that microbiological contamination is higher in dry sand rather than water or sands from superficial water [7,29,35]. In line with the above findings, our results showed the significantly higher colonization of sand (either dry or wet) with several species of fungi. This may have happened due to the affinity of microorganisms with special niches in the sediment, instead of overlying water. Our results demonstrated that potentially harmful fungi were isolated in 67.54% of all the samples, among which species of the genus *Aspergillus* constituted the highest proportion of the fungal species. Species of the *A.* section *Flavi*, especially *A. flavus***,** have been described as the predominant *Aspergillus* species and one of the major etiological fungal agents causing either colonization or allergic/infectious respiratory diseases in Iran [36,37,38,39]. Therefore, people exposed to high amounts of conidia may have an enhanced risk of developing respiratory symptoms. Moreover, increased trends in emerging azole-resistant *Aspergillus* species make disease management more complicated [40,41,42,43]. According to our results, 10.5% and 2.6% of *Aspergillus* strains showed high MICs even against PSZ and ISZ, respectively. Although dermatophyte species were isolated in 7.2% of the samples, the presence of dermatophytes, both associated with humans and animals, can be a great concern, especially for individuals participating in sand activities, which increase the risk of contact and transmission of conidia. In beachgoers with predisposing factors such as immune suppression or diabetes, which make them vulnerable to fungal infections, exposure to the conidia of potentially pathogenic fungi would be of great risk [44,45]. Hence, the sanitary management of beaches can play an important role regarding public health protection. It is worth noting that WHO (2021) does not recommend the disinfection of sand but, rather, the proper management of the coastline. According to the most recent recommendations of WHO, pollution sources for beach sand should be included in sanitary surveys, such as animal excreta, including that of dogs, birds, and other locally significant animals, or human feces. Moreover, management strategies for beaches include the suitable design of solid waste disposal facilities, provision of toilet facilities, and appropriate stormwater drainage [3]. On the other hand, beachgoers should be informed about and encouraged to exercise high-quality personal as well as public hygiene and be educated to follow policies such as feeding wildlife and the disposal of trash. In conclusion, pathogenic fungal species were detected in 67.54% of all samples collected from both water and sand on the Caspian Sea coastline, comprising species that present a hazard to public health. Hence, the monitoring of the health quality of beaches must be looked upon as a relevant public protection aspect, as explained in the new WHO guidelines [3] and pointed out in WHO’s fungal priority pathogens list, to guide research, development, and public health action [5].

## 5. Conclusions

According to the recently released WHO document, species of the genera *Aspergillus*, *Candida*, and *Fusarium* are categorized as high-priority fungal groups. Although these species are commonly found in the environment, multiple exposure to these fungal elements would be a hazard according to the importance of Caspian Sea coastlines for recreational activities in Iran. Although the association with human infection through recreational water exposure is unclear, swimming was found to be a risk factor for otomycosis and for keratitis while wearing contact lenses [46]. Moreover, we should seek to increase the awareness of beachgoers regarding simple hygiene procedures, such as avoiding leaving garbage in beach areas or personal health issues. The preparation of standard sanitation programs, the awareness of beachgoers, and the establishment of criteria to evaluate risks to public health are crucial in the health management of such locales.

## Figures and Tables

**Figure 1 ijerph-20-00459-f001:**
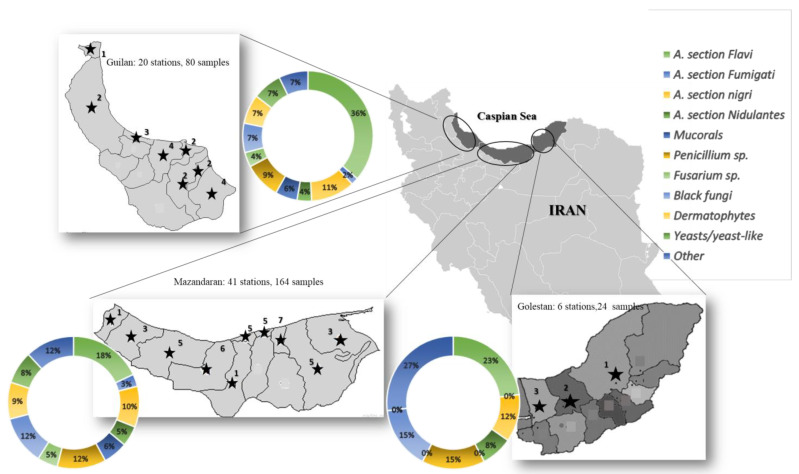
Distribution of the stations and isolated species in the Caspian coastline stations regarding geographical borders (Mazandarn, Guilan, and Golestan provinces). Each (★) indicates cities and the numbers represents the number of stations for sampling in each city.

**Table 1 ijerph-20-00459-t001:** Sequences of primers used in this study.

Fungal Strain	Primer	Sequence (5′–3′)	Related Restriction Enzyme	Reference
*Aspergillus* sp.	Bt-F Bt-R	GGTAACCAAATCGGTGCTGCTTTC- ACCCTCAGTGTAGTGACCCTTGGC-	*Alw*1	[23]
Dermatophytes	ITS-F ITS-R	GCACCTTCAGTCGTAGAGACG- GCACCTTCAGTCGTAGAGACG-	*Mva*1	[25]
Yeast sp.	ITS-F ITS-R	GCACCTTCAGTCGTAGAGACG- GCACCTTCAGTCGTAGAGACG-	*Msp*1	[24]

**Table 2 ijerph-20-00459-t002:** Detailed data for identified isolates according to the sample type.

		Number of Isolates Collected from Each Sample			
	Number of Fungal Species	Dry Sand	Wet Sand	Water Sand	Water
Hyaline filamentous fungi	*Aspergillus* section *Flavi* (44)	15	18	5	6
	*Aspergillus* section *Nigri* (19)	7	5	3	4
	*Aspergiilus* section *Fumigati* (4)	3	1	0	0
	*Aspergillus* section *Nidulantes* (9)	2	5	2	0
	*Penicillium* sp. (21)	7	5	4	5
	*Mucor* sp. (5)	2	3	0	0
	*Rhizopus* sp. (4)	1	2	0	1
	*Trichoderma* (18)	5	2	5	6
	*Fusarium* sp. (7)	2	4	1	0
	*Geothricom* sp. (2)	0	2	0	0
	*Acromonium* (3)	2	1	0	0
Black filamentous fungi	*Cladosporium* sp. (5)	3	2	0	0
	*Bipolaris* (2)	1	1	0	0
	*Alternaria* (3)	3	0	0	0
	Other black fungi (10)	4	3	1	2
Yeast/yeast-like fungi	*Thrichosporon* (3)	2	1	0	0
	*Candida* sp. (5)	3	1	0	1
	Other yeast species (4)	2	1	0	1
Dermatophytes	*Trichophyton mentagrophytes/interdigitale* (10)	6	3	0	1
	*Microsporum canis* (1)	1	0	0	0
	*Microsporum gypseum* (2)	1	1	0	0
	Total (181)	72	61	21	27

**Table 3 ijerph-20-00459-t003:** Antifungal susceptibility profile of *Aspergillus* species isolated from mycosands.

*Aspergillus* Isolate			** Antifungal Agent					
			AMB	VRZ	ITZ	PSZ	ISZ	EFZ
*Aspergillus* section *Flavi* (*n* = 44)	* MICs (µg/mL)	MIC50	0.5	0.25	0.25	0.25	0.5	0.25
		MIC90	0.5	0.5	2	1	2	2
		GM	0.3372	0.2704	0.3267	0.3067	0.3161	**0.1684**
		MIC range		0.125–1	0.062–2	0.06–1	0.062–2	0.031–2
*Aspergiilus* section *Nigri* (*n* = 19)		MIC50	0.125	0.5	0.25	0.5	0.5	0.5
		MIC90	0.5	2	2	0.5	2	4
		GM	0.1798	0.5785	**0.3471**	0.6943	**0.3594**	0.6000
		MIC range	0.031–0.5	0.25–2	0.125–2	0.25–4	0.031–4	0.125–4
*Aspergiilus* section *Nidulantes* (*n* = 9)		MIC50	0.5	0.25	0.25	0.125	0.062	0.125
		MIC90	1	0.5	2	0.5	0.5	0.25
		GM	0.6299	0.25	0.3149	0.1573	**0.0986**	0.1348
		MIC range	0.25–1	0.125–0.5	0.125–2	0.062–0.5	0.062–0.5	0.031–0.25
*Aspergiilus* section *Fumigati* (*n* = 4)		MIC50	ND ^♣^	ND	ND	ND	ND	ND
		MIC90	ND	ND	ND	ND	ND	ND
		GM	1	0.4994	0.5946	**0.2494**	0.4989	0.5
		MIC range	0.5–8	0.062–4	0.125–8	0.062–0.5	0.062–1	0.125–4
*Candida* sp. (*n* = 5)		MIC50	ND	ND	ND	ND	ND	-
		MIC90	ND	ND	ND	ND	ND	
		GM	0.6771	0.1764	0.25	0.6898	0.0364	
		MIC range	0.125–2	0.031–1	0.062–1	0.125–4	0.016–1	

* MIC: minimum inhibitory concentration; ** Antifungal agents: AMB: Amphotricine B, ITZ: Itraconazole, VRZ: Voriconazple, PSZ: Posaconazole, ISZ: Isavaconazole, EFZ: Efinaconazole, FLZ: Fluconazole; ^♣^ ND: Not defined due to the low number of strains. The most effective antifungals for each section of *Aspergillus* was indicated in bold format.

## Data Availability

Some or all data related to AFST and species identification that support the findings of this study are available from the corresponding author (M.N.) upon reasonable request.

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
