# Peer review of "Caspian Sea Mycosands: The Variety and Abundance of Medically Important Fungi in Beach Sand and Water"

_ijerph, 2022, doi:10.3390/ijerph20010459_

Round 1
Reviewer 1 Report (Previous Reviewer 1)
The manuscript has been improved. Some points still need attention.
1) The susceptibility profiles are more relevant for dermatophytes in this scenario than Aspergillus for example. Please add susceptibility info at least for the dermatophytes.
2) Some italics still missing.
Author Response
Dear Editor, dear reviewer
About point 1, the authors would like to state the following:
"The authors greatly appreciate the concern of the reviewer, especially in light of recently emerging reports of anti-fungal resistance in this group of fungi (dermatophytes) but dermatophytes cause superficial infections and that assessment would clinically only make sense in the case of an infection. Aspergilli and Candida were the only ones tested in this project following to the international concern to fight invasive fungal infections correctly from the start. These infections have very high mortality rates and there are currently many known resistances to the species tested. An attempt to characterize the local beach flora on these taxa was the driving force of the anti-fungal testing performed in this study. Dermatophytosis treatments also require a different set of antifungals which are very host/infecton-dependent and more lipophilic"
Point 2: Thank you very much for pointing it out! Hopefully, they are now all italicised.
This manuscript is a resubmission of an earlier submission. The following is a list of the peer review reports and author responses from that submission.
Round 1
Reviewer 1 Report
The study described the variety and abundance of medically important fungi in sand and water samples collected from The Caspian Sea. The study is relevant and interesting. The introductions needs improvements and the objective needs to be better defined. The authors defined “the distribution of fungi collected from Caspian Sea beach sands” as the main objective. But it was not possible to define such distribution at the end of the study. It was not possible because the sampling analyses done for such objective were very low. In addition I might guess that some of the medically important fungi monitored here (e.g. Aspergillus) can be found anywhere. I suggest the redefine the study objectives. The title was clearer as it mentioned the description of the variety and abundance of fungi, much easier to achieve than its spatial distribution.
A second point that needs to be clear in the introduction is the difference between contamination and colonization. Fungi colonize all soils and therefore it can also colonize sand and water. It is very different from “contamination” that refers to the presence of a fungi in a sample where it is not expected to be found or should not be found. Along the discussion, the authors mention (e.g. in line 243 “…significant higher contamination of sand…”) sand and water contamination. What evidence do they have of such contamination? For some fungi, I might accept the term but for Aspergillus. It is very difficult to agree with the expression “Aspergillus contamination” in sand samples just because a single conidia was found there.
The discussion needs improvements. The most relevant points were not really discussed in my opinion (more details below).
Additional comments:
1) Section 2.1. Does a minimal distance was considered for sample collection?
2) Were samples collected from places without people or avoiding the presence of people for at least 24h? Otherwise, how can the authors avoid such association with human presence and human skin (some of these fungi are present on human skin)?
3) Multiple italics are missing along the text. “sp.” should be correctly written along the manuscript.
4) Susceptibility profile. The susceptibility profile of all 181 fungi should be presented here. The number of isolates is not high and therefore all information is valuable.
5) Section 3.2. wild type classification: references are missing.
6) Figure 2 can be removed as the information is a duplicate of what is presented in table 2.
7) Discussion, Line 229. The presence of pets is a very important factor. No information on pet presence is mentioned regarding the beaches tested here.
8) Line 261. “Hence, sanitary management of beaches can play an important role…” Which actions can be made on this regard? What do the authors propose? Does this concept apply to all fungi described in this study? It may be hard regarding Aspergillus and Candida for example.
9) Regarding point 8, abundance of each fungi is a critical point. Can we in fact have abundance values regarding the fungi described here? What levels can be considered acceptable or risky? This I think is the true discussion to be done on the topic. The simple presence of Aspergillus, Penicillium or Candida, for example, in sand or water samples can be considered normal in my opinion. But should the abundance values be taken with caution when certain levels are observed? More information on this topic should be given in the discussion.
Author Response
Please see word attached

Reviewer 2 Report
The paper, titled "Caspian Sea Mycosands: “The Variety and Abundance of Medically Important Fungi in Beach Sand and Water" provides data for the first time on the fungal content of sand and water along the Caspian Sea in Iran. This is a simply and clearly written paper with very valuable data on the fungal community in a given area. Considering the results presented in this paper that potentially pathogenic hyaline fungi were identified in 75% of the samples, the paper confirms the importance of including fungi in the World Health Organization (WHO) guidelines for recreational water quality assessment. It will be a good basis for further research in this area and for raising awareness of the importance of identification and detection of fungi in monitoring the quality of bathing water and sand on beaches.
General
Please make sure that all Latin names in the text are written in italics and with a capital initial letter, including the figures.
Materials and methods
Lines 97-103: I suggest to reorder the sentences in the paragraph, for example:
“The sampling event occurred at the peak of the summer (highest concentration of users) from 1st June to 15th September 2021. The samples were collected only in sunny days between 11 a.m. to 15 p.m. when there were slight temperature changes. Ten grams of sand and 10 mL of water samples were collected with sterile gloves into a sterile plastic container. Each sand sample was collected by combining the sub-samples from the corners and the center of a 2*2 square. Samples were then transported to the laboratory and processed within 24 h.”
I also recommend to explain the sample collecting procedure clearer. Did you take the sand samples at the corners and the center of a square and then take the same quantity (10 grams) of each subsample to prepare one sample? Please explain.
Lines 120-121: Although the cited references are freely available, a brief description of the procedures with basic information would be useful and appropriate.
Results
Lines 206-212: Part of the figure 1 and 2 captions (the second sentences) should probably be removed to the results? Please see the comments in the PDF document.
Conclusion
Lines 271-273: Consider whether you need this section at all. You can include the sentence from the conclusion as the last sentence in the Discussion section.
References
Please use abbreviated journal names instead of full names throughout the reference section. Please see https://www.mdpi.com/journal/ijerph/instructions
Some Latin names are not italicized, please check the references (e.g. 10, 37, 42, 44).

Author Response
Please see the word file attached
